# Relationship between Acylsugars and Leaf Trichomes: Mediators of Pest Resistance in Tomato

**DOI:** 10.3390/insects13080738

**Published:** 2022-08-17

**Authors:** Renato Barros de Lima Filho, Juliano Tadeu Vilela Resende, João Ronaldo Freitas de Oliveira, Cristiane Nardi, Paulo Roberto Silva, Caroline Rech, Luiz Vitor Barbosa Oliveira, Maurício Ursi Ventura, André Luiz Biscaia Ribeiro Silva

**Affiliations:** 1Departament of Agronomy, Universidade Estadual do Centro-Oeste do Paraná–UNICENTRO, Alameda Élio Antonio Dalla Vecchia Street, 838-Vila Carli, Guarapuava 85040-167, Paraná, Brazil; 2Departament of Agronomy, Universidade Estadual de Londrina–UEL, Celso Garcia Cid Roadway (PR-445), Km 380, Londrina 86057-970, Paraná, Brazil; 3Department of Horticulture, Auburn University, 124 Funchess Hall, Auburn, AL 36849, USA

**Keywords:** *Solanum pennellii*, plant breeding, whitefly, allelochemical

## Abstract

**Simple Summary:**

Several pests have the potential to cause major damage to tomatoes. The two-spotted spider mite and whitefly are examples of pests that attack tomato crops. Management of these pests involves several techniques; the use of chemical products is the most adopted strategy. However, growing resistant genotypes is a promising strategy in pest management, reducing the use of chemical products. In this work, we sought to identify genotypes more resistant to the mite and whitefly using advanced populations of tomato (F_2_BC_3_) obtained from the cross between *S. lycopersicum* “Redenção” and *S. pennellii*, accession LA-716. Results indicated a greater resistance in the genotypes with high levels of acylsugars, a chemical compound commonly found in *S. pennellii* tomatoes. Consequently, genotypes with increased levels of acylsugar can be used in breeding programs for pest resistance.

**Abstract:**

Tomato plants are highly susceptible to pests. Among the control methods, genetic improvement with introgression of resistance genes from wild accessions into commercial tomato lines is the best alternative for an integrated pest management (IPM). Thus, the objective of this study was to select tomato genotypes in advanced populations (F_2_BC_3_), with higher levels of acylsugar content, greater recurrent parent genome recovery, and resistance to *Tetranychus urticae* and *Bemisia tabaci* inherited from *Solanum pennellii*. For pest resistance, bioassays were assessed: nine high-acylsugar genotypes, four low-acylsugar genotypes, and the parents, *Solanum lycopersicum* or ‘Redenção’, and *Solanum pennellii* LA-716. Glandular and non-glandular trichomes were quantified. A negative correlation was measured between acylsugar content in the leaflets and pest behavior. Pest resistance was found in the selected F_2_BC_3_ genotypes with high-acylsugar content, indicating that this allelochemical was efficient in controlling the arthropod pests.

## 1. Introduction

The tomato crop (*Solanum lycopersicum*) is very susceptible to insect and disease pressures, which leads to significant yield losses if not properly managed during crop development. Consequently, the development of an IPM strategy that can efficiently control key pests is required to ensure grower profitably. Among the main pests that threaten the crop, the whitefly (*Bemisia tabaci* Gennadius, 1889 (Hemiptera: Aleyrodidae)) and the two-spotted spider mite (*Tetranychus urticae* Koch, 1836 (Acari: Tetranychidae)) are considered the most devastating pests for tomato production due to their direct and indirect damage caused to the crop [1,2,3,4,5].

The application of chemical insecticides is the most common approach adopted to control pests in tomato plants; however, heavy spraying programs have caused economic, ecological, and health problems. Among the issues, inappropriate utilization of chemical insecticides can: increase the resistance of arthropod pests, by selecting occasional advantageous mutated individuals [1,2]; reduce the population of natural enemies; increase chemical residues in fruits; and poison farmers and consumers if not properly managed [3,4]. Therefore, to reduce the dependence on insecticides, research using alternative management methods has been conducted, with emphasis on the development of resistant or tolerant tomatoes [5]. The selection of plants resistant to arthropod pests, obtained through interspecific crosses of elite tomatoes and wild tomato species, followed by backcrossing to the cultivated species, is a promising strategy to obtain pest-resistant plants by introgression of genes of interest [6,7].

*Solanum pennellii* is one of the most used tomato wild species used for crossbreeding focused on pest resistance [8]. *S. pennellii* has elevated contents of acylsugars and a broad-spectrum mechanism of action against arthropod pests, such as *Bemisia tabaci* (Hemiptera: Aleyrodidae), *Tuta absoluta* (Lepidoptera: Gelechiidae), *Tetranychus urticae* (Acari: Tetranychidae), *Diabrotica sapeciosa* (Coleoptera: Chrysomelidae), and other pests [9,10,11]. This allelochemical has a deleterious effect on pest development, reduces oviposition, and decreases the attractiveness of plants to arthropod pests through an antixenosis and antibiosis-type resistance mechanism [12]. The resistance trait can be transferred to commercial tomatoes, as previously reported in the literature [9,12]. Those studies have utilized segregating high-acylsugar genotypes in bioassays and showed negative correlation between the concentration of acylsugars and the biology and behavior of pest arthropods. Nevertheless, it is not always an easy task to achieve tomato genotypes that combine good levels of resistance and desirable commercial traits, mainly because the acylsugars content is a fairly complex trait controlled by 3 or 4 quantitative trait loci (QTLs) [13,14]. For this reason, the use of molecular markers to identify genotypes with greater recovery of *S. lycopersicum* background, containing high-acylsugar content and higher resistance to arthropod pests, becomes a fundamental tool in genetic improvement [9,12] and advancement of breeding programs.

In general, pest-resistant tomato genotypes may reduce pesticide use and its environmental impacts, improving farmer and consumer quality of life, as well as enabling more sustainable and competitive agricultural systems. Thus, the objective of this study was to select tomato genotypes in advanced populations (F_2_BC_3_), with higher levels of acylsugar content, greater recurrent parent genome recovery, and resistance to *Tetranychus urticae* and *Bemisia tabaci* inherited from *Solanum pennellii*.

## 2. Materials and Methods

### 2.1. Experimental Plant Material

In this study, F_2_BC_3_ genotypes developed and selected by the tomato breeding program of UEL/UNICENTRO were used. To develop these genotypes, the commercial tomato ‘Redenção’ (*S. lycopersicum*) was crossed with the wild tomato *S. pennellii* (LA-716). *S. pennellii* was obtained from the University of California in 1997, and plants have been maintained by the coordinator of the research group of this breeding program. After obtaining the F_1_ hybrids, successive backcrosses and selections were conducted until obtaining the F_2_BC_2_ genotypes [15].

The selected plants from the F_2_BC_2_ population (RVTA-2010-31-177-pl#39; RVTA-2010-31-177-pl#177, and RVTA-2010-31-319-pl#214) were pollen donors in backcrossing with the recurrent parent ‘Redenção’, generating the F_1_BC_3_ population. With the self-fertilization of individuals from the F_1_BC_3_ generation, we obtained individuals from the F_2_BC_3_ population. The F_2_BC_3_ genotypes were sown in 128-cell expanded polystyrene trays containing commercial substrate based on biostabilized pine bark and expanded vermiculite. Trays were kept in a greenhouse with an average temperature of 25 ± 3 °C and relative humidity of 78 ± 4% for germination and seedling formation. Seedlings were irrigated by micro-sprinklers. Plants were transplanted at the 3–4 leaf stage into 5 dm^3^ polypropylene pots containing sieved sub-surface soil (10–40 cm depth), corrected with 5 g of calcium carbonate, and fertilized with 30 g of N-P-K (4-14-8), according to recommendations for the crop and soil chemical analysis.

Acylsugar content in leaflets was determined according to the methodology proposed by Resende et al. [8,16] in 550 F_2_BC_3_ plants and in 20 plants of each parent, used as controls for the identification of low- and high-acylsugar genotypes. Forty-five days after transplantation, six 1 cm diameter leaf discs totaling 4.21 cm^2^ of leaflet area were collected from the upper portion of plants and placed in test tubes with 1 mL dichloromethane for extraction of acylsugars. The tubes were stirred with a Vortex mixer for 30 s. Discs were removed, the solvent was evaporated, and 0.5 mL 0.1 N sodium hydroxide dissolved in methanol was added. The mixture was evaporated, the residue was maintained at 100 °C and methanol was added three times at 2-min intervals to guarantee the completion of the saponification reaction. After evaporation of the methanol, the residue was dissolved in 0.4 mL water.

Acylsugars can exist as both acylglucose and acylsucrose; therefore, after saponification a mixture of glucose and sucrose is obtained. Sucrose was converted into glucose and fructose by adding 0.1 mL 0.04 N hydrochloric acid, boiling for 5 min and cooling. The Sommogy–Nelson reagent (Nelson, 1944) was added, and the mixture was heated to boiling for 10 min and cooled to room temperature in a stream of cold water. Arsenomolybdate (0.5 mL) was added, the solution was stirred on a Vortex mixer for 15 s and the absorbance was measured at 540 nm in a spectrophotometer (UV-VIS Cary 60) (Nelson, 1944).

The concentration of acylsugars is directly proportional to the absorbance, so samples with higher absorbance values have the higher levels of this allelochemical [8,16]. This procedure was performed in four repetitions for each treatment (plant). The pre-selected plants were subjected to a second round of analysis in two replications in order to validate the acylsugar levels.

The selected plants were divided into two groups (i.e., high and low acylsugar). Eighteen genotypes were selected for the high-acylsugar group and four genotypes were selected for the low-acylsugar group (Table 1). Plants of those genotype were transferred to 10-dm^3^ pots containing commercial substrate based on biostabilized pine bark, expanded vermiculite, and subsurface soil. From these plants, axillary shoots were taken for vegetative propagation by cuttings. The cuttings were rooted in 72-cell trays containing the same substrate mentioned above and then transferred to 7-dm^3^ pots, where they rooted.

### 2.2. Evaluation of Recurrent Parent Genome Recovery

Recurrent parent genome recovery was determined on the eighteen high-acylsugar F_2_BC_3_ genotypes selected in the previous step (RVTA pl#13, RVTA pl#24, RVTA pl#32, RVTA pl#61, RVTA, pl#88, RVTA pl#127, RVTA pl#180, RVTA pl#192, RVTA pl#230, RVTA pl#238, RVTA pl#253, RVTA pl#292, RVTA pl#306, RVTA pl#325, RVTA pl# 374, RVTA pl# 376, RVTA pl#402, RVTA pl# 440, and RVTA pl#511), and the parents *S. lycopersicum* ‘Redenção’ and *S. pennellii* LA-716.

For DNA extraction, young leaflets from the apical region of plants were used, following the existing methodology [17]. Plant material was ground in liquid nitrogen and aliquots of 100 mg of tissue were separated in microtubes and kept at −20 °C until extraction. One mL of the extraction buffer (20 mM EDTA, 100 mM Tris-HCl (pH 8.0), 2 mM NaCl, 2% CTAB, 2% PVP, 2% β-mercaptoethanol) was added to each microtube containing the plant tissue. Subsequently, tubes were incubated in a water bath at 65 °C for 30 min. Phenol: chloroform: isoamyl alcohol (25:24:1) was added, and the DNA was separated from the solution by centrifugation, precipitated with ethanol in the presence of Polyethylene Glycol (PEG). To obtain a high degree of purity, the material was successively washed with ethanol. Ultimately, the DNA was resuspended in TE (10 mM Tris-HCl (pH 8.0), 1 mM EDTA) with RNase, incubated at 37 °C for 30 min, and stored in a freezer at −20 °C.

DNA quantification was performed through agarose gel (0.8%) electrophoresis, stained with ethidium bromide, and visualized under ultraviolet (UV) light. Phage λ DNA was used to estimate DNA concentration in the samples.

### 2.3. PCR-ISSR

Sixteen ISSR primers were used in the PCR reactions (Table 2). The reactions were prepared with a final volume of 12.5 µL containing 20 ng DNA, 0.2 µM of primer, 200 µM of each dNTP, 1.5 mM of MgCl_2_, 1U of Taq DNA Polymerase, and 1× of PCR buffer [18,19].

The thermal cycler was programmed with initial DNA denaturation at 94 °C for 5 min, followed by 35 cycles at 94 °C for 45 s, primer annealing temperature (Table 2) for 45 s, and 72 °C for 90 s. Then, a final extension step was performed at 72 °C for 5 min. The amplification products were revealed by agarose gel (1.8%) electrophoresis, stained with ethidium bromide (0.5 µg mL^−1^), run at a constant current of 110 volts for four hours, and visualized under UV light. The 100 bp DNA ladder was used to estimate the size of the fragments. The ISSR primers and the PCR amplification protocol used in this study were evaluated and considered efficient in tomato by Resende et al. (2021).

### 2.4. Quantification of Glandular and Non-Glandular Trichomes

Of the 18 genotypes with high content of acylsugars in the laboratory, nine were identified for estimating the density of leaf trichomes based on marker-assisted selection, which identified those with the highest proportion of the recurrent parent. For identification and quantification of trichomes, leaflets were collected from the upper portion of nine genotypes pre-selected in the previous phase, presenting greater recurrent parent genome recovery and high-acylsugar content (RVTA pl#24H, RVTA pl#32-H, RVTA pl#61-H, RVTA pl#180-H, RVTA pl#230-H, RVTA pl#238-H, RVTA pl#292-H, RVTA pl#306-H, and RVTA pl#325-H), along with four low-acylsugar genotypes (RVTA pl#17-L, RVTA pl#42-L, RVTA pl#69-L, and RVTA pl#123-L), and the pest-resistant (*S. pennellii*) and pest-susceptible (‘Redenção’) controls.

For each leaflet, approximately 25 mm^2^ were cut by a scalpel blade, and then the samples were fixed on a sample holder (stubs) with carbon tape [20] and observed under a scanning electron microscope (SEM-VEGA3 TESCAN). Ten 1 mm^2^ images of the abaxial and adaxial sides of all repetitions, were taken at 200× magnification. From the 100 images obtained for each genotype, the 50 betters were selected for quantification of the density of glandular and non-glandular trichomes present in the leaflets according to the scheme proposed by Luckwill [21]. For analysis, the trichomes were separated into glandular and non-glandular.

### 2.5. Resistance Bioassays for Two-Spotted Spider Mite

#### 2.5.1. No-Choice Traveled Distance Bioassay

The bioassay was conducted according to the existing methodology adapted [22]. A young leaflet of each genotype with the abaxial surface facing upwards was fixed in the center of a sheet of white paper by thumbtack, placed on a polystyrene plate. Ten adult (12-day-old) females of *T. urticae* were released on the head of each thumbtack. After 20, 40, and 60 min, the distance traveled by the mites (mm) on the surface of each leaflet was measured using a millimeter ruler. For the mites that remained on the thumbtack (without moving), the distance traveled was considered zero, and for the mites that left the leaflet surface, towards the paper sheet, the distance from the thumbtack and the farthest extremity of the leaf lamina was considered (Figure 1A).

The test was performed in an air-conditioned room with temperature of 20 ± 3 °C and RH 70 ± 4, in a completely randomized design with 20 replicates. The statistical analysis considered the mean distance traveled by the ten mites in each repetition.

#### 2.5.2. No-Choice Oviposition Bioassay

The experiment was conducted in sample units mounted in 50 mm diameter Petri dishes, lined with a sponge saturated with distilled water and a thin layer of moistened cotton (Figure 1B). One leaf disk with 15 mm diameter (1.76 cm^2^), obtained from the median portion of 50-day-old tomato plants, was placed in the center of the plate. Using a stereoscopic microscope (Olympus^®^ SZ61, Tokyo, Japan) and a fine bristle brush, six adult (12 days after hatching) females were released.

The experiment was conducted in a climate-controlled chamber with temperature of 25 ± 3 °C, relative humidity 70 ± 10%, and photophase of 12 h, in a completely randomized design with 20 repetitions. Evaluations were performed 24 h after the establishment of the experiment, using a stereoscopic microscope (Olympus^®^ SZ61) to count live mites and eggs.

### 2.6. Resistance Bioassay to Whitefly Bemisia Tabaci

The free-choice bioassay was conducted in a greenhouse in a completely randomized design, with 15 genotypes and four repetitions. The treatments consisted of nine high-acylsugar genotypes, four low-acylsugar genotypes, and the pest-susceptible (‘Redenção’) and pest-resistant (*S. pennellii* LA-716) controls (mentioned above). The trial was conducted in a completely randomized design, with four replications, in which plots consisted of a 5 dm^3^ polypropylene pot, filled with pine bark substrate, containing one tomato plant of each genetic material, containing one tomato plant of each genetic material. These pots were kept in a greenhouse (6.00 m [w] × 6.00 m [l] × 3.80 m [h]), with a temperature of 25 ± 4 °C and relative humidity of 75 ± 10%. For whitefly rearing, common bean plants (*Phaseolus vulgaris*) were kept in cages with anti-aphid screen which received whitefly adults collected in an experimental sweet potato field.

For laboratory creation of whitefly, vases (5 dm^3^) of beans at the R1 stage, were used and kept in the rearing cages for 48 h. After that, the pots and the adults were removed, letting only eggs remain in the trifolium of the bean plants.

Artificial infestation of whiteflies in the greenhouse was performed according to the existing methodology adapted [23]. For this, pots with host plants containing eggs were placed between every two tomato plants, 50 cm apart. The plants were infested in the pre-flowering stage, 45 days after transplantation.

Evaluations were performed at 10, 17, 24, and 31 days after infestation by counting eggs and nymphs present in one cm^2^ of leaf area. Two leaflets from the lower, middle, and upper portion of the plants were sampled. Counting was performed using a stereoscopic microscope (Olympus^®^ SZ61). The oviposition preference index (OPI) was calculated by the formula OPI = [(T − C)/(T + C)] × 100, where T is the number of eggs observed in the evaluated treatment (genotype) and C is the number of eggs counted in the susceptible control, ‘Redenção’.

### 2.7. Statistical Analysis

Concerning the molecular data, the gels were visually evaluated according to the presence (1) or absence (0) of bands. With the matrix formed, a similarity analysis was performed between *S. lycopersicum* ‘Redenção’ and the evaluated genotypes. The genetic similarity was estimated by Jaccard’s coefficient using the NTSYS 2.2 (NY-USA) software.

Data regarding acylsugar content and mite bioassays were subjected to the Kruskal–Wallis test, followed by the Dunn–Bonferroni pairwise multiple-comparison test. The results obtained in the whitefly bioassay were subjected to the Friedman test, with subsequent grouping based on the Fisher criterion. Spearman correlation was performed to verify the influence of acylsugars on the results obtained. Orthogonal contrast tests were also carried out to compare the groups of genotypes using the Scheffé test, parents were not used to establish the groups, they were used only to compare with high or low acylsugar. The hierarchical clustering analysis was performed, considering the data obtained from the acylsugar content and the bioassays with the evaluated pests. Trichome count data were compared through the Kruskal–Wallis test, followed by the Dunn–Bonferroni test. The Kruskal–Wallis test, Friedman test, Scheffé test, correlations, and hierarchical clustering were performed in the Rstudio statistical program using the agricolae [24] and dendextend [25] packages.

## 3. Results

High-acylsugar genotypes were considered those with absorbances above 0.300 nm [15]. Based on this value, eighteen and four high- and low-acylsugar genotypes, respectively, were selected. Low-acylsugar genotypes were used in the whitefly and two-spotted spider mite resistance bioassays as comparison criteria (Table 3). Average absorbance values obtained in *S. pennellii* were higher than those in the other treatments. Genotypes with high-acylsugar content did not differ among themselves; however, they differed from those with low-acylsugar content, including ‘Redenção’.

The ISSR primers amplified 46 polymorphic loci that were used in the genetic similarity analysis. Genotypes, even within the group selected for high levels of acylsugars, presented different levels of genetic similarity with the recurrent parent (Figure 2). From the eighteen genotypes evaluated for similarity with the recurrent parent, nine were selected, considering those with similarity values above 36% (Figure 2). The genotypes RVTA pl#32 (73) and RVTA pl#325 (73) had the most genetic similarity to the recurrent parent (Figure 2).

The nine genotypes with the highest genetic similarity to the recurrent parent, identified through assisted selection by ISSR markers, and the four genotypes with low content of the allelochemical and the resistant (LA-716) and susceptible (‘Redenção’) controls were analyzed for density of glandular and non-glandular trichomes on the leaflets (Figure 3A–D). Table 4 shows the differences measured among the tomato genotypes evaluated for these traits. Particularly, *S. pennellii* LA-716 had the highest density of glandular trichomes (Figure 3D) on both leaflet faces (abaxial and adaxial). The genotypes RVTA pl# 180-H and RVTA pl# 306-H (Figure 3B) presented the closest values to the glandular trichome densities of *S. pennelli*.

Genotypes grouped as high acylsugar had no significant differences to those grouped as low-acylsugar in relation to the density of glandular trichomes. All high-acylsugar genotypes showed higher density of glandular trichomes than ‘Redenção’. As for the non-glandular trichomes, the highest density occurred on the abaxial face. Neither the allelochemical content correlated significantly with trichome density nor total glandular trichomes (Table 4). Correlations and contrasts were not significant for the characters acylsugar content and types of leaf trichomes.

Differences among the genotypes evaluated were identified for the average distances traveled by mites in all evaluation times (Table 5), in which *S. pennellii* LA-716 had the lowest traveled distance by mites that all other genotypes, regardless of the evaluation time. Particularly, after 20 and 40 min of mite release, traveled distance was shorter on the abaxial surface of the leaflets in the high-acylsugar genotypes compared with the low- acylsugar genotypes and the recurrent susceptible parent, ‘Redenção’ (Table 5). After 60 min of the mites’ release, the greater traveled distance was measured in the low-acylsugar genotypes. Results indicated that the high-acylsugar genotypes were more effective in reducing the movement of mites. However, RVTA pl#61-H and RVTA pl# 292-H did not differ from the low-acylsugar RVTA pl#69-L. Among the high-acylsugar genotypes, RVTA pl#24H, RVTA pl#32H, RVTA pl#180H, RVTA pl#238, RVTA pl#306 RVTA pl#325, and RVTA pl#230H stood out, differing from the low-acylsugar genotypes and ‘Redenção’ (Table 5).

Correlations between the acylsugar content in the leaflets and the average distance traveled by mites (Table 5) demonstrated the effect of the allelochemical on increasing mite resistance. Significant and negative correlations of −0.698, −0.722, and −0.715 were found at 20, 40, and 60 min, respectively. The C1 contrast (Table 5) significantly highlighted that the high-acylsugar group was more effective in reducing traveled distance by mites than the low-acylsugar group. The C2 contrast compared the results from the high-acylsugar genotypes and ‘Redenção’, in which, the former had lower distances traveled by mites compared with the susceptible control. The acylsugars present in the high-content genotypes were more effective, increasing mite resistance compared with the low-content genotypes.

In *S. pennellii* LA-716 (resistant control), mite population had lower survival compared with those maintained on other genotypes (Figure 4). The low-acylsugar genotypes such as RVTA pl#17-H and the susceptible control ‘Redenção’, provided higher survival the female mites compared with the high content. Some high-acylsugar genotypes did not differ from low-acylsugar genotypes concerning mite adult survival. The accession ‘LA-716’ presented the lowest level of mite oviposition. All the high-acylsugar genotypes differed from the low-acylsugar genotypes and the susceptible control for oviposition (Figure 4). 

The correlation between acylsugar content and number of live mites, although moderate, was negative and significant (−0.475 **). However, a strong correlation was measured between acylsugar content and number of eggs on leaflets (−0.731 **) (Table 6). The C1 contrast, which compares the high- and low-acylsugar genotypes, was significant for the number of eggs. As well as the comparison of the number of eggs in the leaflets between the genotypes “Redenção” and “high acylsugar” (contrast C2) denotes significant results, indicating a negative influence on the oviposition of the mites caused by the high-acylsugar content (Table 6).

Tested genotypes influenced the preference of whitefly for oviposition, as well the development of nymphs, regardless of the evaluation time (Table 7). *S. pennellii* LA-716 had the greatest interference in whitefly behavior for the evaluated traits. Among the genotypes selected for the high-acylsugar group, some had significant responses for all traits, during the evaluation points (i.e., RVTA pl#180-H, RVTA pl# 230-H, RVTA pl#238-H, RVTA pl#292-H, RVTA pl#306-H, and RVTA pl#325-H). Three other high-acylsugar genotypes evaluated were unstable regarding resistance level. The genotypes RVTA pl#24, RVTA pl#61, and RVTA pl# 32 had no impact on whitefly oviposition in the initial three evaluations; however, they significantly impacted the development of nymphs (Table 7). It is emphasized that even though these genotypes did not differ from the low-acylsugar genotypes in some evaluations, they presented higher values than the susceptible control. The fourth evaluation demonstrated more clearly this separation of contrasting genotypes, in which all high-acylsugar genotypes differed from the low-acylsugar genotypes for oviposition and nymph development, except RVTA pl#24.

The oviposition preference index (OPI) evinces the effectiveness of the high-acylsugar genotypes in reducing whitefly oviposition, probably by the non-preference mechanism of resistance (Table 7). For instance, the genotypes RVTA pl#230, RVTA pl#238, RVTA pl#292, RVTA pl#306, and RVTA pl#325 showed OPI more than 50% lower than the susceptible control. A significant and negative correlation was measured between the acylsugar content in the leaflets with all the variables evaluated, indicating that the whitefly behavior was directly influenced by acylsugars, that is, when acylsugar concentration increases, the number of eggs and nymphs present in the leaflets decreases (Table 7).

Results obtained from the proposed contrasts led to the identification of differences among the groups tested. For instance, when comparing the genotypes selected for high-acylsugar group with those of low content (C1), differences between genotypes were noted. For all variables, it was measured that the genotypes in the high-acylsugar group were less oviposited and presented less nymph development. The genotypes in the low-acylsugar group had greater values for oviposition and were not able to reduce the number of whitefly nymphs effectively, especially compared with *S. pennellii* LA-716 (Table 7). When the high-acylsugar genotypes were compared with ‘Redenção’ (C2), differences were observed among them. The average number of eggs and nymphs observed in the susceptible control was higher, evidencing the existence of components of resistance to whitefly in the high-acylsugar content genotypes (Table 7).

In the comparison between the low-content genotypes and ‘Redenção’ significant differences were only found for the number of nymphs, in the evaluation of 7 days, and number of eggs in the upper portion of leaflets at 17 days. In the other evaluations, the low-content genotypes behaved similarly to the susceptible control (Table 7).

In the hierarchical clustering analysis, two clusters were formed. Group one (red) gathered ‘Redenção’ and genotypes with low-acylsugar content, and group two was composed of S. pennellii LA-716 and genotypes with high-acylsugar content. Within group two (high-acylsugar genotypes), two other groups were formed: the first (green) grouped, besides S. pennellii, the genotypes RVTA pl#230, RVTA pl#292, RVTA pl#306, RVTA pl#238, and RVTA pl#325; and the second (blue) grouped RVTA pl#32, RVTA pl#24, RVTA pl#61, and RVTA pl#180 (Figure 5).

Figure 5 also shows that, although S. pennellii LA-716 is grouped with high-content genotypes, it is more distant from the others and the resistance levels presented by the genotypes, based on the evaluated characters.

## 4. Discussion

The presence of acylsugar in advanced generation genotypes obtained from the cross between commercial tomatoes and the wild accession *S. pennellii* LA-716 guarantees resistance to several tomato pests [6,7,9]. Results obtained in this study confirmed that the presence of acylsugars in the leaflets of tomato genotypes exerts an influence on biological aspects of *T. urticae* (Figure 4 and Table 5). Although the selected high-content genotypes did not show the same level of resistance found in *S. pennellii*, they showed less susceptibility than the low-content genotypes and the susceptible control, ‘Redenção’. Thus, the F_2_BC_3_ plants that contain higher concentrations of this allelochemical have desirable resistance traits (Table 5, Table 6 and Table 7 and Figure 4). The presence of this allelochemical in tomato leaflets is controlled by QTLs, which regulate the amount and type of acylsugars, composed of acylsucrose and/or acylglucose. The greater or lesser level of them determines the higher or lower level of resistance to certain pests [13,14].

The presence of this allelochemical directly affects the behavior of the two-spotted spider mite, reducing its traveled distance and oviposition (Figure 4 and Table 5). Therefore, tomato lines with higher concentrations of acylsugars negatively affect the traveled distance of mites [26]. Similarly reported was that in which high-acylsugar genotypes decreased the number of eggs deposited on leaflets, increased egg incubation period, reduced egg viability, and reduced nymph survival and choice of *T. urticae* [12]. This allelochemical promotes antixenosis- and antibiosis-type resistance, negatively influencing biological and behavioral aspects of *T. urticae* [12].

For whitefly, it was measured that acylsugar affects the pest behavior, interfering in its choices since lower oviposition preference and lower number of nymphs were found in high-acylsugar genotypes compared with genotypes in the low-acylsugar group (Table 7). When studying the behavior of *Bactericera cockerelli* (Hemiptera: Triozidae) in tomato lines with high-acylsugar content, lower oviposition and higher nymph mortality were observed compared with commercial tomatoes [10]. Additionally, authors reported that the presence of acylsugar in tomato leaflets negatively interferes with *B. tabaci* preference compared with commercial tomato lines [7]. In general, plants with higher concentrations of acylsugars affect the behavior of whitefly. Thus, in crop areas where plants have lower preference for oviposition, the insect population is also be lower, facilitating pest control using alternative management techniques. By observing the negative correlations (Table 7) between number of eggs and nymphs present in the leaflets of tomato plants containing higher levels of acylsugars, it is inferred that there is an antixenosis-type resistance factor [6,10].

Results measured in the pest bioassays indicate that tomato plants with higher levels of acylsugar have great potential for transferring this resistance trait when generations are advanced through backcrossing and they also suggest that these resistance-providing traits are heritable in each generation of backcrossing. In addition, all genotypes classified in the high-acylsugar group influenced the behavior of the pests tested, with genotypes RVTA pl#292, RVTA pl#230, RVTA pl#238, RVTA pl#306, and RVTA pl#325 being more effective. As observed in the hierarchical clustering analysis (Figure 5), these genotypes, together with *S. pennellii* LA-716, form a group, being the ones with the highest acylsugar content and more resistant to pests, compared with the low-acylsugar genotypes and ‘Redenção’. This separation occurred as a result of the levels of acylsugars observed in the selected plants, associated with glandular trichomes and the joint effects on the traits related to resistance to the pest arthropods evaluated.

Genotypes RVTA pl#24, RVTA pl#32, RVTA pl#61, and RVTA pl#180, even being classified in the high-acylsugar group, are not as effective in pest resistance as the resistant control, this can be associated with the type and proportion of acylsugars present in the leaves. This fact explains that, even when we select tomato genotypes with high-acylsugar content, resistance levels are variable. This may be due to the levels of acylsugars, morphological aspects, and the interaction between them [13]. Additionally, there may be other components that promote resistance to arthropod pests, but which were not completely inherited by the offspring during the advancement of generations through backcrossing. Additionally, when the effect of acylglucose and acylsucrose isolated from *S. pennellii* LA-1376 was studied, a different efficacy of the compounds on whitefly and *Frankliniella* spp. (Thysanoptera: Thripidae) was reported [27]. Nevertheless, none of the isolated compounds provided the same efficiency against the pest as when they acted together. This proves that acylsugar types interact synergistically, interfering in pest behavior and favoring the plant. This fact explains that, even when we select tomato genotypes with high-acylsugar content, the amount and proportions of acylglucose and acylsucrose are different from those observed in *S. pennellii*-LA-716; consequently, plants have different resistance levels.

As aforementioned, the resistance level observed in the accession ‘LA-716’ is conferred by acylsugars, which are related to the presence of glandular trichomes on leaves [28,29]. Nevertheless, the presence of acylsugar in the genotypes studied in this work had no correlation with the presence of glandular trichomes (Table 4). Particularly, no correlation between acylsugar and glandular trichomes has been reported, suggesting that leaf structures, other than glandular trichomes such as epidermal layer surfaces may be sites of stored acylsugar or synthesis. No scientific evidence of other structures involved in the synthesis of this allelochemical has been presented so far [30,31].

Among the various possibilities of combination between the concentration of the studied allelochemical and proportion of recurrent parent genome recovery, the ideal plant for advancement in backcrossing is the one that combines high concentrations of this allelochemical and high recovery of the recurrent parent genome (Figure 2). A segregating population can present different allelochemical contents as well as different proportions of the recurrent parent genome. Thus, biochemical tests and biological resistance assays are effective in identifying plants with high allelochemical content. Nevertheless, they do not discriminate plants with higher proportion of the recurrent genome. The use of molecular markers enables the differentiation of genotypes by using similarity coefficients, bringing faster results that facilitate the selection process in recurrent backcrossing [18,32]. In the present study, considering the similarity values along with the results from the bioassays, it was possible to select plants that combine higher pest resistance and higher recurrent parent genome recovery.

Considering the clustering formed in the hierarchical clustering analysis (Figure 5) and the results of genetic similarity (Figure 2), the plants that presented the best results in the pest experiments and had greater genetic similarity with ‘Redenção’ were selected for generation advancement. The genotype RVTA pl#325 showed the highest similarity (73%), followed by RVTA pl#230 (53%) and RVTA pl#238 (51%), with intermediate values. These last-mentioned genotypes represent the ones able to provide greater gain in the advancement of generation since they keep characteristics of pest resistance as well as greater similarity to ‘Redenção’, favoring the selection of a plant with commercially desired traits.

Even though no correlation was found between trichomes and acylsugar content, a negative correlation occurred for acylsugar and the parameters evaluated in the pest resistance bioassays for all behavioral traits. Thus, the genotypes selected from the F_2_BC_3_ population, RVTA pl#292, RVTA pl#230, RVTA pl#238, RVTA pl#306, and RVTA pl#325, with high-acylsugar content and derived from ‘Redenção’ or *S. pennellii* LA-716, are less susceptible to the two-spotted spider mite and whitefly. In addition, among these genotypes, RVTA pl#325 has the highest similarity with the recurrent parent, whereas RVTA pl#230 and RVTA pl#238 present intermediate similarity, being selected to be pollen donors for the next F_1_BC_4_ backcross generation. Furthermore, the development of promising genotypes that may reduce pesticide use in tomato fields is extremely relevant when seeking rational and sustainable crop production systems.

## Figures and Tables

**Figure 1 insects-13-00738-f001:**
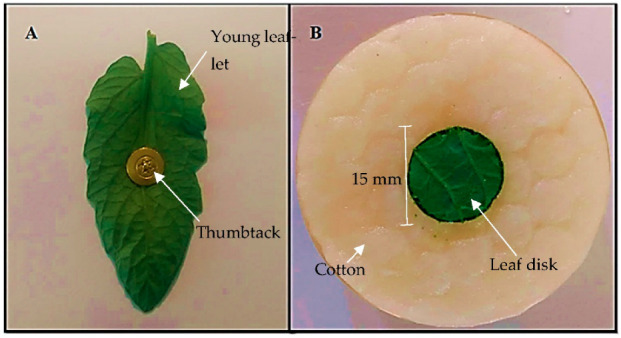
Experimental unit for no-choice, (**A**) traveled distance, and (**B**) oviposition bioassays.

**Figure 2 insects-13-00738-f002:**
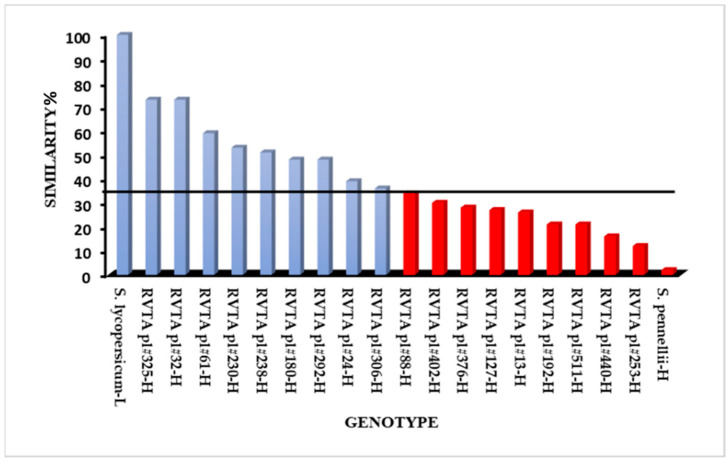
Genetic similarity (%) based on DNA marker analysis of tomato genotypes with high levels of acylsugar, selected from the F_2_BC_3_ population, with the recurrent parent *S. lycopersicum* (‘Redenção’). The horizontal line represents the cutoff point with a similarity index superior to 36%.

**Figure 3 insects-13-00738-f003:**
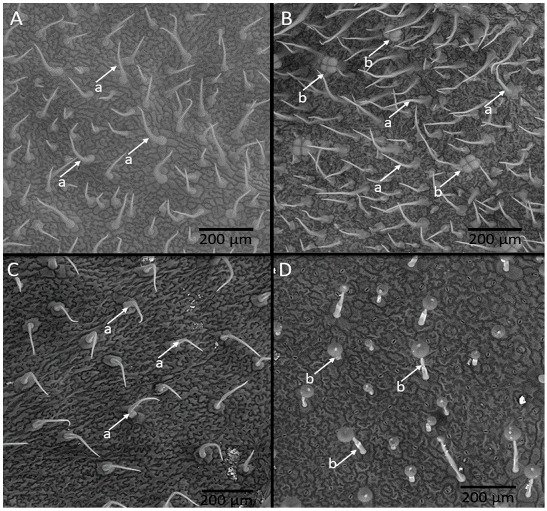
Non-glandular (a) and glandular (b) trichomes present on the surface of RVTA pl#17-Low (**A**), RVTA pl#306-High (**B**), *S. lycopersicum* ‘Redenção’ (**C**), and *S. pennellii* (**D**).

**Figure 4 insects-13-00738-f004:**
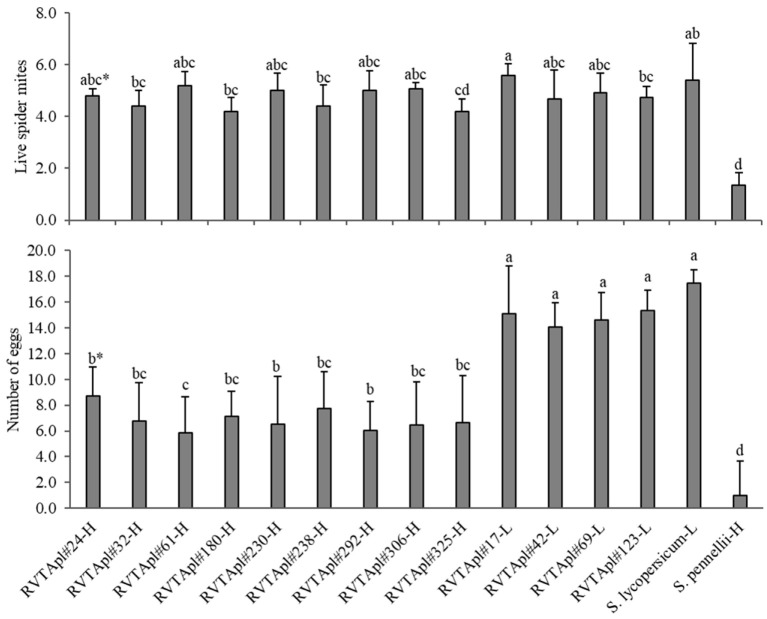
Average number of live mites and eggs in leaf discs of F_2_BC_3_ tomato genotypes 24 h after the release of 6 females of *Tetranychus urticae*. * Averages followed by the same letter do not differ statistically by the Dunn–Bonferroni test, *p* < 0.05; Kruskal–Wallis number of mites: 70.54 *; number of eggs: 173.19 * (*p* < 0.001).

**Figure 5 insects-13-00738-f005:**
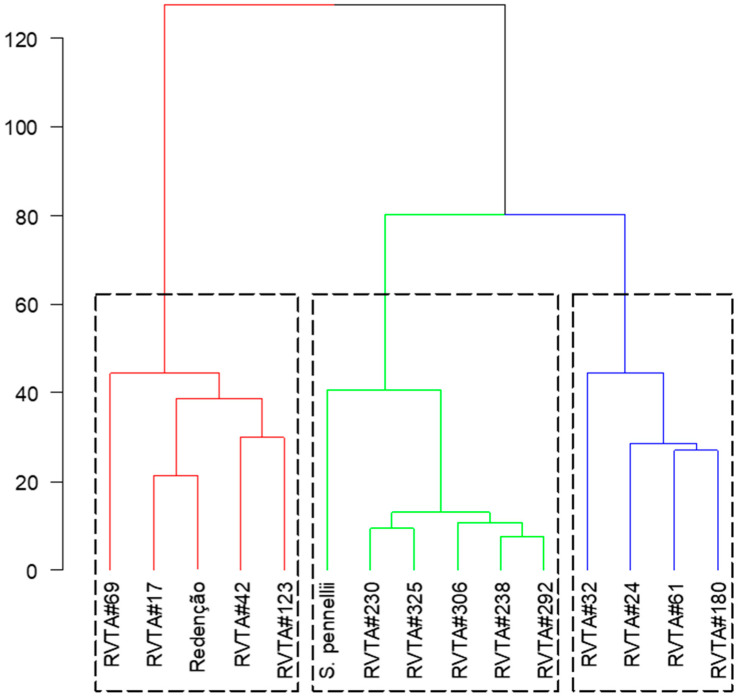
Hierarchical clustering of tomato genotypes, considering acylsugar content and mites and whiteflies behavior in choice and no-choice bioassays.

**Table 1 insects-13-00738-t001:** List of the selected genotypes divided into two groups.

Genotype Pedigree	Genotype Abbreviation	Acylsugar Group ^1^ (abs 540 nm)
RVTA-2010-31-177-39-pl#13	RVTA-pl#13	High
RVTA-2010-31-177-39-pl#24	RVTA-pl#24	High
RVTA-2010-31-177-39-pl#32	RVTA-pl#32	High
RVTA-2010-31-177-39-pl#61	RVTA-pl#61	High
RVTA-2010-31-177-39-pl#88	RVTA-pl#88	High
RVTA-2010-31-177-39-pl#127	RVTA-pl#127	High
RVTA-2010-31-177-39-pl#180	RVTA-pl#180	High
RVTA-2010-31-177-39-pl#192	RVTA-pl#192	High
RVTA-2010-31-177-39-pl#230	RVTA-pl#230	High
RVTA-2010-31-319-214-pl#238	RVTA-pl#238	High
RVTA-2010-31-177-39-pl#253	RVTA-pl#253	High
RVTA-2010-31-177-177-pl#292	RVTA-pl#292	High
RVTA-2010-31-319-214-pl#306	RVTA-pl#306	High
RVTA-2010-31-177-177-pl#325	RVTA-pl#325	High
RVTA-2010-31-177-39-pl#374	RVTA-pl#374	High
RVTA-2010-31-177-39-pl#376	RVTA-pl#376	High
RVTA-2010-31-177-214-pl#402	RVTA-pl#402	High
RVTA-2010-31-319-177-pl#440	RVTA-pl#440	High
RVTA-2010-31-177-214-pl#511	RVTA-pl#511	High
RVTA-2010-31-177-39-pl#17	RVTA-pl#17	Low
RVTA-2010-31-177-39-pl#42	RVTA-pl#42	Low
RVTA-2010-31-319-214-pl#69	RVTA-pl#69	Low
RVTA-2010-31-177-39-pl#123	RVTA-pl#123	Low

^1^ level of acylsugars based on absorbance values.

**Table 2 insects-13-00738-t002:** ISSR primers used to evaluate recurrent parent genome recovery in F_2_BC_3_ genotypes. AT: annealing temperature.

Primers	Sequence (5′-3′)	AT (°C)
UBC 807	(AG)_8_T	52°
UBC 808	(AG)_8_C	50°
UBC 809	(AG)_8_T	51°
UBC 810	(GA)_8_T	52°
UBC 811	(GA)_8_C	52°
UBC 815	(CT)_8_G	52°
UBC 827	(AC)_8_G	53°
UBC 835	(AG)_8_YC	54°
UBC 836	(AG)_8_Y^ *^A	53°
UBC 848	(CA)_8_AGG	55°
UBC 855	(AC)_8_CTT	55°
UBC 864	(ATG)_6_	50°
UBC 866	C(TCC)_5_TC	55°
UBC 873	(GACA)_4_	50°
UBC 890	VHV(GT) _7_	54°

**Table 3 insects-13-00738-t003:** Mean (±standard deviation) values of acylsugar contents at 540 nm levels obtained from contrasting F_2_BC_3_ genotypes (H = high content and L = low content).

Genotype	Acylsugar ^1,2^ (abs 540 nm)
RVTA pl#13-H	0.33 ± 0.07 b ^3^
RVTA pl#24-H	0.41 ± 0.05 b
RVTA pl#32-H	0.44 ± 0.05 b
RVTA pl#61-H	0.39 ± 0.03 b
RVTA pl#88-H	0.41 ± 0.04 b
RVTA pl#127-H	0.40 ± 0.06 b
RVTA pl#180-H	0.33 ± 0.03 b
RVTA pl#192-H	0.31 ± 0.05 b
RVTA pl#230-H	0.38 ± 0.03 b
RVTA pl#238-H	0.43 ± 0.02 b
RVTA pl#253-H	0.34 ± 0.06 b
RVTA pl#292-H	0.32 ± 0.02 b
RVTA pl#306-H	0.32 ± 0.04 b
RVTA pl#325-H	0.33 ± 0.01 b
RVTA pl#376-H	0.32 ± 0.04 b
RVTA pl#402-H	0.39 ± 0.04 b
RVTA pl#440-H	0.36 ± 0.05 b
RVTA pl#511-H	0.36 ± 0.06 b
RVTA pl#17-L	0.05 ± 0.03 d
RVTA pl#42-L	0.16 ± 0.02 c
RVTA pl#69-L	0.12 ± 0.04 cd
RVTA pl#123-L	0.08 ± 0.05 cd
*S. lycopersicum*	0.12 ± 0.02 cd
*S. pennellii*	0.58 ± 0.02 a

^1^ Acylsugar contents at 540 nm. ^2^ The contents of acylsugars are considered from the absorbance values. ^3^ Means followed by the same letter do not present significant statistical differences by the Dunn–Bonferroni test (5%).

**Table 4 insects-13-00738-t004:** Mean number of glandular and non-glandular trichomes (per mm^2^) on the adaxial and abaxial surface of leaflets of contrasting tomato genotypes, selected by the acylsugar content in the F2BC3 population and correlations between the content (abs) and characters related to the types of trichomes.

Genotype	Acylsugar ^1^ (abs 540 nm)	Abaxial Surface (mm^−2^)		Adaxial Surface (mm^−2^)		Total (mm^−2^)	
		Glandular	Non-Glandular	Glandular	Non-Glandular	Glandular	Non-Glandular
RVTA pl#24 (High)	0.41 b *	5.2 b *	53.0 ab *	2.3 b *	121.5 ab *	7.5 cd *	174.5 ab *
RVTA pl#32 (High)	0.44 b	7.2 ab	41.5 a–d	1.2 bc	82.0 de	8.5 c	123.5 d–f
RVTA pl#61 (High)	0.39 b	5.0 b	34.5 de	2.0 bc	71.2 ef	7.0 de	105.7 fg
RVTA pl#180 (High)	0.33 b	8.5 ab	35.5 de	2.5 ab	95.7 cd	11.1 ab	131.2 cd
RVTA pl#230 (High)	0.38 b	5.7 b	45.7 ab	1.7 bc	125.7 ab	7.5 cd	171.5 ab
RVTA pl#238 (High)	0.43 b	4.5 bc	30.0 ef	1.2 bc	76.2 e	5.7 d	106.2 e–g
RVTA pl#292 (High)	0.32 b	4.0 bc	14.5 fg	1.3 c	69.0 e–g	5.3 d	83.5 g–i
RVTA pl#306 (High)	0.32 b	9.0 ab	54.2 a	3.2 ab	117.7 a–c	12.2 ab	172.0 ab
RVTA pl#325 (High)	0.33 b	6.2 bc	20.5 fg	1.2 c	107.0 bc	7.4 cd	127.5 c–e
RVTA pl#17 (Low)	0.05 d	5.2 b	40 b–d	0.0 d	148.5 a	5.2 d	188.5 a
RVTA pl#42 (Low)	0.16 c	1.0 d	37.7 b–d	0.2 d	111.2 bc	1.2 f	149.0 bc
RVTA pl#69 (Low)	0.12 cd	1.2 d	20.0 ef	1.4 bc	74.5 ef	3.0 e	94.5 gh
RVTA pl#123 (Low)	0.08 cd	6.5 ab	34.7 de	1 c	83.0 de	7.5 cd	117.7 d–f
Redenção (Low)	0.12 cd	0.5 d	6.2 g	0 d	20.0 fg	0.5 f	26.2 hi
*S. pennellii* LA-716 (High)	0.58 a	21.5 a	0.0 g	24.2 a	0.5 g	45.7 a	0.5 i

* Averages followed by the same letter do not differ by the Dunn–Bonferroni test (*p* < 0.001); Kruskal–Wallis (H) 61.51. ^1^ Acylsugar contents at 540 nm (based on absorbance values).

**Table 5 insects-13-00738-t005:** Average (±standard deviation) distance covered by mites (*Tetranychus urticae*) at 20, 40, and 60 min after release of females on the abaxial surface of leaflets and correlation between acylsugar content and parameters analyzed in the no-choice behavioral test with F_2_BC_3_ genotypes, performed at 20 ± 5 °C.

Genotype	Acylsugar ^1^ (abs 540 nm)	Traveled Distance (mm)		
		20 min	40 min	60 min
RVTA pl#24 (high)	0.41 b *	11.8 ± 1.3 b *	13.8 ± 1.5 b *	15.7 ± 1.0 b *
RVTA pl#32 (high)	0.44 b	12.7 ± 1.0 b	15.6 ± 0.8 b	16.5 ± 1.1 b–d
RVTA pl#61 (high)	0.39 b	12.9 ± 0.9 b	19.2 ± 0.7 cd	20.2 ± 0.7 d–f
RVTA pl#180 (high)	0.33 b	13.1 ± 1.8 b	13.6 ± 0.5 b	16.0 ± 0.8 bc
RVTA pl#230 (high)	0.38 b	13.1 ± 1.8 b	14.2 ± 0.4 b	18.2 ± 1.5 b–e
RVTA pl#238 (high)	0.43 b	11.5 ± 0.7 b	17.4 ± 0.7 bc	18.9 ± 0.5 b–e
RVTA pl#292 (high)	0.32 b	9.6 ± 0.93 b	15.1 ± 1.61 bc	19.2 ± 0.7 b–e
RVTA pl#306 (high)	0.32 b	14.4 ± 0.4 b	16.4 ± 1.51 bc	18.9 ± 0.9 b–e
RVTA pl#325 (high)	0.33 b	13.6 ± 1.6 b	16.9 ± 0.92 bc	20.8 ± 1.2 c–e
RVTA pl#17 (low)	0.05 d	21.5 ± 2.0 c	25.4 ± 1.64 e	26.6 ± 1.0 g
RVTA pl#42 (low)	0.16 c	19.1 ± 1.7 c	21.6 ± 1.3 de	22.8 ± 1.2 fg
RVTA pl#69 (low)	0.12 cd	20.1 ± 0.6 c	22.2 ± 1.0 de	24.2 ± 1.0 e–g
RVTA pl#123 (low)	0.08 cd	20.1 ± 2.2 c	21.9 ± 1.6 de	24.5 ± 1.3 g
Redenção (low)	0.12 cd	21.8 ± 1.3 c	24.6 ± 0.3 e	25.0 ± 0.5 g
*S. pennellii* LA-716 (high)	0.58 a	0.2 ± 0.2 a	1.2 ± 0.5 a	1.2 ± 0.5 a
H (Kruskal–Wallis)	-	72.5 **	73.8 **	72.5 **
Correlation (r) with acylsugar	-	−0.7 **	−0.7 **	−0.7 **
**Contrasts (C)**				
C1–Genotypes with high acylsugar vs. low acylsugar		−7.8 ^†^	−6.9 ^†^	−6.4 ^†^
C2–Genotypes with high acylsugar vs. ‘Redenção’		−9.7 ^†^	−8.7 ^†^	−6.5 ^†^

* Means followed by the same letter do not differ from each other by the Dunn–Bonferroni test (*p* <0.001); ** Kruskal–Wallis test (*p* < 0.001). Contrasts: ^†^ Significant by Student test at 5% probability. ^1^ Acylsugar contents at 540 nm (based on absorbance values).

**Table 6 insects-13-00738-t006:** Correlation between acylsugar content (based on absorbance at 540 nm) and parameters analyzed in the no-choice behavioral test with F_2_BC_3_ genotypes, performed at 20 ± 5 °C.

Genotype	N° Spider Mite	N° Eggs
Correlation (r) with acylsugar	−0.47 **	−0.73 **
**Contrasts ^1^ (C)**		
C1–Genotypes with high acylsugar vs. low acylsugar	−0.28	−7.88 *
C2–Genotypes with high acylsugar vs. ‘Redenção’	−0.70	−10.58 *

** Spearman’s correlation (p < 0.001). ^1^ Contrasts: * Significant by Student test at 5% probability.

**Table 7 insects-13-00738-t007:** Acylsugar content (based on absorbance at 540 nm) and average number of eggs and nymphs at 7, 14, 21, and 28 days after infestation (DAI) of whitefly in F_2_BC_3_ tomato genotypes and oviposition preference index (OPI).

Genotype	Acylsugar ^1,2^ (abs 540 nm)		1st Evaluation 10 DAI				2nd Evaluation 17 DAI				3rd Evaluation 28 DAI				4th Evaluation 31 DAI				OPI ^1^
			Eggs cm^−2^		Nymphs cm^−2^		Eggs cm^−2^		Nymphs cm^−2^		Eggs cm^−2^		Nymphs cm^−2^		Eggs cm^−2^		Nymphs cm^−2^		
RVTA pl#24 (High)	0.41	b *	20.2	ab *	0.0	d *	39.5	c–e *	10.5	de *	37.0	e *	29.2	bc *	42.7	d *	34.75	c *	−30.03
RVTA pl#32 (High)	0.44	b	21.7	ab	0.0	d	20.5	d–f	9.2	cd	51.0	b	34.7	ab	67.2	c	39.75	bc	−23.53
RVTA pl#61 (High)	0.39	b	9.0	cd	0.0	d	26.0	e–g	17.2	bc	34.5	e	28.7	c	50.7	cd	14.00	d	−36.63
RVTA pl#180 (High)	0.33	b	10.6	c	0.0	d	22.2	f–h	16.0	cd	36.5	de	18.2	d	31.2	d	12.75	d	−44.08
RVTA pl#230 (High)	0.38	b	7.5	cd	0.1	cd	11.0	Ij	7.5	d–f	12.7	fg	8.5	ef	23.2	e	13.00	d	−65.26
RVTA pl#238 (High)	0.43	b	9.4	cd	0.2	cd	14.0	hi	4.5	fg	11.0	fg	15.5	d	23.0	e	7.50	de	−63.73
RVTA pl#292 (High)	0.32	b	6.1	c–e	0.0	d	17.5	hi	6.7	ef	10.0	gh	16.7	de	18.0	e	7.25	ef	−66.82
RVTA pl#306 (High)	0.32	b	10.1	cd	0.0	d	17.2	g–i	9.2	d–f	19.0	f	13.5	de	19.0	e	5.75	ef	−59.69
RVTA pl#325 (High)	0.33	b	4.1	de	0.9	bc	10.2	ij	7.0	ef	17.7	f	10.2	ef	16.7	ef	9.25	de	−68.28
RVTA pl#17 (Low)	0.05	d	24.2	ab	7.7	a	78.7	a	43.5	a	93.7	a	40.2	ab	126.0	a	48.75	ab	10.91
RVTA pl#42 (Low)	0.16	c	30.5	ab	4.5	b	46.7	bc	31.0	ab	43.2	cd	28.2	bc	82.0	ab	34.75	bc	−12.28
RVTA pl#69 (Low)	0.12	cd	18.9	b	2.2	bc	34.2	d–f	30.7	ab	51.7	bc	31.5	bc	90.2	ab	29.50	c	−14.10
RVTA pl#123 (Low)	0.08	cd	22.7	ab	6.0	ab	57.7	a–c	41.0	a	62.0	b	46.2	a	68.5	b	56.50	a	−10.51
*S. lycopersicum* Redenção (Low)	0.12	cd	31.7	a	10.0	a	68.0	ab	43.0	a	71.0	ab	45.2	a	88.5	ab	51.00	ef	0.00
*S. pennellii* LA-716 (High)	0.58	a	0.0	e	0.0	d	0.0	j	0.0	g	0.2	h	0.0	f	0.0	f	0.00	f	−99.81
**Correlation (r)**	-		−0.6 ^a^		−0.6 ^a^		−0.7 ^a^		−0.9 ^a^		−0.6 ^a^		−0.5 ^a^		−0.6 ^a^		−0.57 ^a^		-
**Contrasts**																			
C1–Genotypes with high acylsugar vs. low acylsugar	-		−9.5 *		−4.2 *		−31.6 *		−21.9 *		30.9 *		−16.1 *		−44.7 *		−20.77 *		-
C2–Genotypes with high acylsugar vs. ‘Redenção’	-		−19.6 *		−9.6 *		−47.3 *		−31.7 *		−43.6 *		−25.4 *		−51.7 *		−33.27 *		-

^1^ OPI = [(T − C)/(T + C)] × 100, where T is the number of eggs observed in the treatments (genotypes) and C is the number of eggs counted in the susceptible control, ‘Redenção’. * Averages followed by the same letter do not differ by the Dunn–Bonferroni test (*p* < 0.001); Kruskal–Wallis (H) 61.51; ^a^ Spearman’s correlation (*p* < 0.001). ^1^ Acylsugar contents at 540 nm. ^2^ Levels of acylsugars based on absorbance values.

## Data Availability

All datasets presented in this study are included in the article and can be provided by the authors upon reasonable request.

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
