# Peer review of "Relationship between Acylsugars and Leaf Trichomes: Mediators of Pest Resistance in Tomato"

_insects, 2022, doi:10.3390/insects13080738_

Round 1
Reviewer 1 Report
Several items are flagged in the manuscript. Please address these comments. Also, because we have known for a long time that sugar esters from Solanum trichomes are related to resistance, this idea should no longer be considered novel. The use of background selection should be applauded, but the data presented in this manuscript just barely scratches the surface, due in part to the extremely small sample size, and the application of background selection to a very limited number of individuals. The background selection info could have been considerably improved if accompanied by phenotypic data, especially with regard to reproduction.

Author Response
We appreciate the two anonymous reviewers’ and return the manuscript after consider all the suggestions and comments.
In order to response each comment made by the reviewers, we describe below our procedures in the second version of the manuscript.
All answers are in the PDF file.

Reviewer 2 Report
In this study, the authors report the screening of a population of tomato for resistance and increased tolerance to biotic stresses. To provide additional support for the role of acylsugars in biotic stress resistance they crossed a high acylsugar producing species (S. pennellii) with a low acylsugar producing species (S. lycopersicum) and following backcrosses analyzed the progeny. First they quantified total acylsugars and grouped the progeny into two classes, low and high acylsugar producing species. They then tested these progeny against different pests and found that lines that accumulated high acylsugars were more pest resistant than the low accumulators. Interestingly, they found that trichome density did not correlate with plant defense, only acylsugar abundance. Overall, this study confirms the findings of previous reports for the roles of acylsugars to an array of biological stresses. The reviewer has no major additional experiments to propose but there are several areas where the authors should provide crucial methodological details and other areas for improvement.
1. The method used for acylsugar quantification is not sufficiently detailed in the methods. The authors also mention Resende et al., 2002 on ln. 101, but this article is not in the references section. A reducing sugar assay is not ideal for quantification of acylsugars as there are many other compounds that could absorb at 550nm and lead to over or underestimation of acylsugar levels. For example, free sugars such as sucrose and glucose could contaminate these measurements. It is not clear if a phase separation was performed first to eliminate polar free sugars, prior to quantifying acylsugars. Also, reducing sugar assays only measure reducing sugars. This would only detect acylglucoses and not acylsucroses (which is a non-reducing sugar). Thus, the authors may have only quantified acylglucose levels and measured the effects of high acylglucose levels on pest interactions. Although a reducing sugar assay is OK for this type of study to examine the differences between high and low levels, the methods need to be clearly explained so the reader can interpret the results fully. It is not clear the selection criteria used to separate high and low acylsugar accumulating genotypes. The authors should comment in the discussion or elsewhere that it can’t be determined from this way of quantification if individual acylsugars were reduced in these genotypes or if total acylsugars were reduced. It may be possible that there are differences in the metabolic pathways for example different ASATs that may alter the levels of an individual acylsugar compared to changes in a transcription factor that controls all of acylsugar biosynthesis that may result in total reduction.
2. Levels of acylsugars did not correlate with trichome density. In pennellii there are abundant type I/IV trichomes (acylsugar producing trichomes), but in the RVTA pl # 306-High there are abundant type VI trichomes. Perhaps there would be differences in correlations if the authors separated into trichome type and not just glandular and non-glandular.
3. It is not clear what Fig 6, the hierarchical clustering is based on. Is this based on the ISSR genotype, effects of insect feedings or acylsugar content. Please clarify in the figure legend or methods.
4. Were other metabolites measured beyond acylsugars? It is possible that in lines that have high acylsugar contents that they also have high accumulation of other defense metabolites like glycoalkaloids or terpenes. The authors should discuss this if not measure for some other metabolites.
5. What do the overall physiology of the plants looks like. If this is to be used as a strategy for breeding an edible tomato with high defenses, do the high-acylsugar lines look like S. lycopersicum and does the fruit quality match that of S. lycopersicum?
6. fix acyl-sugars to acylsugars. There are many instances throughout the text.
7. In the summary portion at the top of the manuscript it is unclear what F2RC3 is.
8. The tables would be more easy to read is the genotypes were sorted from low to high or high to low acylsugar contents instead of numerical order.
9. Some of the figures are missing axis labels, e.g., Fig 5 mission y-axis label. It is not clear in figure 5 what the height of green and yellow bars are. Again, this might be better visualized if genotypes are sorted left to right by acylsugar levels and not numerically.
10. What does AAs content in Table 4 refer to?
11. The lines in the discussion 472-474 that discuss potential additional sites of acylsugar synthesis in the mesophyll cells seems a bit unlikely, given the extreme trichome enrichment of acylsugar biosynthetic genes in the trichomes. It is more likely that greater metabolic flux through existing trichomes is giving rise to increased acylsugar levels in some genotypes. This portion of the discussion also sites papers that don’t seem to be relevant to acylsugar biosynthesis and I recommend citing other papers here.
Author Response

(The authors gave the same response as above.)

Round 2
Reviewer 1 Report
I stopped reading the manuscript when I discovered Figure 2 on line 448 which comes after Figure 5 on line 368. There are too many introduced errors in this manuscript to permit me to go forward. There are extraneous figures still in the manuscript (the old incomprehensible Fig 2) and both color and b&w renditions of Fig 4. Figure and table call outs in the text do not match the actual table. I have attached a file that has some of my suggested edits for the ms.

Author Response
Dear reviewer,
We appreciate the anonymous reviewer and return the manuscript after considering all the suggestions and comments.
Pg 1, line 12: ISSR markers were not used to identify genotypes with high acylsugar content. A test for sugar was used for this. Markers were used for background selection.
R: This sentence is wrong, we exclude it.
Pg 02, line 49: Subject verb agreement
R: Adjusted
Pg 02, line 31: This phrase is not a new item. It modifies item i.
R:
Pg 02, line 61: This is patently untrue. Several other wild species have been used.
R: Adjusted.
Pg 02, line 90: State origin of this line, including year, and maintenance in subsequent years.
R: Adjusted. S. pennellii was obtained from the University of California in 1997, being used in the dissertation and doctoral work of the coordinator of this breeding program. The seeds were provided by Dr. Charles Rick.
Pg 02, line 95: Unclear. Were individuals selfed, or was the entire population sib-mated via bulk pollination? Be precise.
R: Adjusted.
Pg 03, line 111: Discs?
R: Yes. Adjusted.
Pg 03, line 125: This needs a reference.
R: Adjusted.
Pg 04, line 148: aliquots, misspelled.
R: Adjusted
Pg 04, line 148: placed, not separated.
R: Adjusted
Pg 05, line 193: for, not to.
R: Adjusted
Pg 06, line 219: Please provide photos with greater clarity. Also, it would be useful to lable the photos (thumbtack, leaf disc, cotton, etc.
R: Adjusted.
Pg 06, line 244: It is unlikely that you created whiteflies.
R: To reduce the error in the age of the whiteflies, we performed the creation with as much control as possible. So, we placed pots with beans, without the presence of insects, in cages where there was whitefly breeding, and 48 hours later we removed them, and kept the adults away from the leaves with slight movements of the plants and checking that there were no more insects on the leaves. A relatively difficult job but done to maintain the standard of the experiment.
Pg 07, line 269: Indicate that these did not include parents.
R: Adjusted.
Pg 07, line 270: Specifically indicate what groups of genotypes were evaluated. Specifically did this single df contrasts include parents?
R: Adjusted. The contrasts were performed from the formation of distinct groups, high, low, susceptibility control and resistance control.
Pg 07, line 279: This is not nanometers (nm). It is likely absorbance units.
R: Adjusted.
Pg 07, line 287: Where is table 3? Missing?
R: Adjusted
Pg 08, line 291: I presume that the old Fig 2 should have been deleted.
R: Adjusted
Pg 09, line 303: Why are there two graphs here?
R: Adjusted. One was exclude.
Pg 10, line 325: Table 4?
R: Adjusted
Pg 10, line 328: This is an undefined abbreviation. What is AAs and what are the units?
R: the acronym AAs refers to 'acylsugars'. We have included information in the legend.
Pg 11, line 331: resistant, not resistance.
R: Adjusted
Pg 11, line 352: Table 5?
R: Adjusted
Pg 11, line 356: What are the units?
R: Adjusted.
Pg 11, line 356: Not an English word. Also, these are probably F-values. Please label them as such.
R: We prefer to delete this word and keep only the word 'contrasts'.
Pg 12, line 365: Accession
R: Adjusted
Pg 12, line 368: Which graphs am I supposed to be looking at? Reduce significant digits on y-axis. Label the two graphs and refer to labels in table legend.
R: Adjusted, only the first graphic is correct, the other one has been deleted.
Pg 12, line 369: italic.
R: Adjusted
Pg 14, line 393: See earlier comments about the use of this word.
R: Adjusted.
Pg 16, line 433: Reduce significant digits
R: We reduce the values to only 1 decimal place, as suggested.
Pg 16, line 435: Put genotypes in order of ABS.
R: Adjusted
Pg 17, line 448: Figure 2??
R: Adjusted
Reviewer 2 Report
All the comments have been addressed.
I do not agree with the statement about acylsugars being biosynthesized in the underlying mesophyll cells. To make such a claim the authors should provide data to support this statement or a reference that would support this claim.
Author Response
Dear Review, We thank the anonymous reviewer and returned the manuscript after considering all suggestions and comments.
Perhaps we made a mistake in writing the sentence, perhaps the acylsugars can be synthesized at sites other than trichomes and be stored in pockets in the cells of the mesophyll, it's still not clear. We understand that the acylsugars present in these pockets are not extracted by the methodology used. This explains some genotypes with low allelochemical content being more tolerant to pests.
Research carried out by Luu, et al., 2017 showed in some of their results that the silencing of genes responsible for the synthesis of O-AS in tobacco reduced between 20 and 30% only the allelochemical. Also in this work, they applied several techniques to wash the leaf surface, removing the O-AS present in the trichomes of the species and proved that the washing reduced the weight gain of M. sexta, but that not all allelochemicals were removed. So, the question arises: Where are the rest of the O-AS being stored? We understand that it may not be possible to say that they are in the mesophyll but scanning microscopy studies have identified pockets with contents that may be acyl sugars in tomato (data not yet published).
Periclinal grafting experiments and isotopic labeling studies using epidermal strips (with trichomes attached) indicate that the epidermal layer sufficient for acyl sugar production in S. pennellii (Goffreda et al., 1990; Kroumova and Wagner, 2003). Direct demonstration of synthesis in detached trichomes of S. pennellii has not been shown, although this is the likely site of synthesis (Wagner, 1991; Kroumova and Wagner, 2003).
It is not known if S. pennellii tomato glandular trichomes are photosynthetically competent, but chlorophyll-containing plastids have been identified in the major trichome types in cultivated tomato (Pike and Howells, 2002). As acyl sugar accumulation can reach very high levels in S. pennellii and in some tobacco species (Fobes et al., 1985; Wagner, 1991), it is likely that trichomes import at least some proportion of the photosynthates they need for acyl sugar production from other leaf tissues.
Round 3
Reviewer 1 Report
There remain some problems with this manuscript. The citation of tables and/or figures that do no exist are particularly problematic and indicate what I interpret as particularly grievous and reflect very poorly on all ten (10!) of the authors which I presume have read each version of this manuscript. And this assumes that the existence of the errors are due to oversight by all ten authors. These sorts of errors cause this reviewer to lose all confidence in the scientific ability of the authors.
The tables must be able to stand alone, and consequently all columns must be clearly labeled and indicate the units represented in the column. To this end, columns labeled "Acylsugar" generally have not included the units in this version of the manuscript. Thus the tables are unacceptable. These columns should be labeled "Absorbance units" or "Optical density" if the latter was determined and probably should include the wavelength. The authors interpret these data as acylsugars, but the actual data is related to light absorbance at a particular wavelength. The tables should so indicate this fact.
Other problems are marked in the manuscript and are mainly grammatical problems.
I checked the bibliographic entries, but did not verify that each entry was cited nor did I check for duplicates.

Author Response
Dear reviewer,
we appreciate the considerations applied during this round.
We fixed the errors that were present in the article and, we made all the suggested changes.
As for the references, we used software to help, but they were later verified.
Reviewer 2 Report
none
Author Response
none
Round 4
Reviewer 1 Report
I apologize for taking additional time for this review. This time of year, during field season, it usually takes me one week to find the time to do a review. On top of that my Adobe Acrobat installation crashed as I was finishing this review requiring a complete reinstallation of Acrobat. I think most of the comments are understandable. Because of the changes to Acrobat, I had to rename the file.

Author Response
Dear Editor,
We appreciate the two anonymous reviewer and return the manuscript after consider all the suggestions and comments.
In order to response each comment made by the reviewrs, we describe below our procedures in the second version of the manuscript.
Pg. 01, line 15: These pests management -> Management of these pests
Response: We rewrite the sentence to improve the concordance.
Pg. 01, line 41: Growers -> Grower
Response: We rewrite
Pg. 02, line 50: , -> ;
Response: We rewrite
Pg. 02, line 53: managements -> management
Response: We rewrite
Pg. 02, line 64: Through, not throughhout
Response: We rewrite
Pg. 03, line 100: 3, not -3
Response: We rewrite
Pg. 03, line 104: tge
Response: We rewrite
Pg. 03, line 112: metanol
Response: Adjusted to methanol
Pg. 03, line 117: As a comment, this solution is likely still basic, based on the amount of NaOH tha was previously added to the sample and that remained after evaporation.
Response: The sodium hydroxide used was consumed in the saponification reaction of the acylated groups present in the sugar molecule, the rest neutralized with acid, according to the methodology of Resende et al. 2002.
Pg. 03, line 119: Arsenomolibdate -> Arsenomolybdate
Response: We rewrite
Pg. 03, line 123: Highest -> higher
Response: We rewrite
Pg. 04, line 159: This procedure needs a reference. Who validated these markers?
Response: Adjusted.
Pg. 05, line 175: The modifying phrase is mis-placed
Response: We modified the phrase
Pg. 05, line 184: In -> For
Response: We rewrite
Pg. 05, line 186: This was impossible to accomplish. One side of cutting was stuck to the stub, so it was impossible to visualize abaxial and adaxial surfaces of each cutting.
Response: Adjusted
Pg. 05, line 189: The scheme proposed.
Response: We rewrite
Pg. 06, line 226: resistant, not resistance.
Response: We rewrite
Pg. 06, line 228: Considerable less detail about pot size and media here in this section, compared to information presented earlier.
Response: We added information about the vase and substrate
Pg. 06, line 230: P. vulgaris = First use of genus, spell out.
Response: We rewrite
Pg. 07, line 253: Spotted = removed
Response: We removed
Pg. 07, table 3: These letters ar undefined in the table
Response: We have included the statistical test used for the comparison of means at the bottom of the table.
Pg. 08, line 288: This was not a DNA analysis. It was a marker analyssis, or DNA marker analysis
Response: We rewrite
Pg. 08; line 288: Between = of not between
Response: We rewrite
Pg. 09, line 308: With acylsugars groups = delete this frase
Response: We deleted this phrase
Pg. 09, line 308: Correlations have nothing to do with orthogonal contrasts. Where are the contrasts?
Response: Adjusted.
Pg. 09, line 310: Does not mention correlation
Response: Adjusted
Pg. 10, line 318: Them -> Than?
Response: We rewrite
Pg. 10, line 325: Acylsugars genotypes = spelling
Response: We rewrite
Pg. 10, line 330: To be consistent, should be past tense
Response: We rewrite
Pg. 10, line 337: Constrasts not mentioned
Response: Adjusted
Pg. 11, line 341: White space
Response: We rewrite
Pg. 11, line 344: The figure shows live mites not dead mite. The figure deals with mite survival, not mute mortality
Response: We rewrite
Pg. 11, line 345: Assuming that you are talking about mite survival, not mortality, the presente results not support this conclusion. Survival on #32, 180, 238, and 325, was same as on three of the four low sugar genotypes
Response: We rewrite
Pg. 11, line 360: Add asterisks for consistency?
Response: We included asterisks
Pg. 11, line 362: These contrasts do not support the stated conclusion.
Response: We added a statement to give more consistency to the sentence regarding contrasts, but we understand it to be a result and not merely a conclusion.
Pg. 12, line 379: It appears to me that the values are mostly lower, not higher.
Response: Is correct. we mean that genotypes with low content showed some resistance to the pest, when compared, for example, to the genotype Redenção. This indicates that there may be other resistance components.
Pg. 12, line 380: This is not true at all.e.g. For nymphs at 31 DAI 32 did not differ from 42 and 17
Response: But that's exactly what we're saying, that some genotypes high in acylsugars didn't differ for nymphs and eggs. And that this is more evident in the fourth evaluation, except for 24.
Pg. 14, line 412: Please label these groups or refer to line colors.
Response: We rewrite
Pg. 14, line 414: What was the underlying cause of these aggregations. Whats were the means of the variables for each group?
Response: Adjusted. The clusters are formed as a result of the evaluated characteristics, in this case the allelochemical content and the resistance levels presented, as described in the results.
Pg. 14, line 420: I think if you get the means for each group, you will discover that Other variables may be involved as well
Response: The hierarchical clustering analysis was performed considering the data obtained from the acylsugar content and all the means obtained in biological and behavior bioassays. In such analyses, we undertake that the set of parameters are considered together to compose the results. In case we use the mean for each group separately, we probably will characterize other variables involved. However, this is not the purpose of this specific analysis.
Pg. 14, line 424: There is not mention in the results with regard to how behavior affected clustering
Response: Cluster analysis does not permit us to separate the effect of each parameter in the arrangement of the groups. Therefore, we can not infer how ‘behavior parameters’, alone, could affect clustering.
Pg. 14, line 437: Only one of them? Based on the literature, it would seem to me that levels of both of them are related to resistance
Response: We removed the expression ‘only’
Pg. 15, line 442: Missing sentence subject
Response: We rewrite
Pg. 15, line 452: Have reported
Response: We rewrite
Pg. 15, line 453: Abbreviate genus?
Response: We rewrite
Pg. 15, line 455: Concentration -> concentrations
Response: We rewrite
Pg. 15, line 461: While this may be true, you did not this in this paper. No mention of how earlier Generation were screened is mentioned anywhere. Youo can not offer such a briad conclusion with not data.
Response: Adjusted
Pg. 15, line 467: See comment about cluster means.
Response: Adjusted
Pg. 15, line 472: You can speculated on this, but since you have no data in this regard, that is all you can do. Modify this discussion accordingly.
Response: That's what we did in the text, we speculate. So I don't see the need to change the wording.
Pg. 15, line 482: A concluison completely unsupported by this research. You did not measure sucrose vs glucose
Response: Adjusted
Pg. 15, line 489: This is not a sentence;
Response: We deleted this sentence.
Pg. 16, line 493: This could also be simply the amount per trichome.
Response: Iit were just trichomes, all correlations would be significant between the structures and the allelochemical content. Some works suggest that other structures present in the mesophyll may be associated.
Pg. 15, line 512: Move this so that it follows “similarity”.
Response: Adjusted
Pg. 16, line 514: Ones that may be able. You did not show this in the paper.
Response: How not? If the genotypic constitution of a genotype is in greater proportion of the cultivated species, in future generations the commercial gain will be greater and faster. That's why we use selection assisted by molecular markers.
Pg. 17, line 567: Is the reference complete?
Response: Yes, this reference is complete.
Pg. 17, line 571: Page renge?
Response: We insert the requested page range.
Pg. 17, line 575: Page range?
Response: We insert the requested page range.
Pg. 17, line 580: Pages?
Response: We insert the requested page range.
Pg. 18, line 618: Is the reference complete?
Response: Yes, this reference is complete.
Pg. 19, line 629: Page range?
Response: We insert the requested page range.
Pg. 19, line 638: Page range?
Response: We insert the requested page range.
Pg. 19, line 644: ??????
Response: We delete the error.

Round 5
Reviewer 1 Report
Editors, please see comments to authors.

Author Response
Dear reviewer,
We appreciate your comments about our work.
We've made the necessary changes and we're coming back.
We are uploading the zip file, with two files, the clean text and another where we made the changes.
Thank you very much.
